# Polyethylene Protective Coating with Anti-Reflective Properties for Silicon Photovoltaic Cells

**DOI:** 10.3390/ma16114004

**Published:** 2023-05-26

**Authors:** Malgorzata Pociask-Bialy

**Affiliations:** College of Natural Sciences, University of Rzeszów, Pigonia 1 Str., 35-959 Rzeszów, Poland; mpociask@ur.edu.pl

**Keywords:** photovoltaic cells, antireflective coatings, solar glass transmittance, short-circuit current gain

## Abstract

The aim of the study was to find the effect of polyethylene (PE) coatings on the short-circuit current of silicon photovoltaic cells covered with glass, in order to improve the short-circuit current of the cells. Various combinations of PE films (thicknesses ranging from 9 to 23 µm, number of layers ranging from two to six) with glasses (greenhouse, float, optiwhite and acrylic glass) were investigated. The best current gain of 4.05% was achieved for the coating combining a 1.5 mm thick acrylic glass with 2 × 12 µm thick PE films. This effect can be related to the formation of an array of micro-wrinkles and micrometer-sized air bubbles with a diameter of 50 to 600 µm in the films, which served as micro-lenses and enhanced light trapping.

## 1. Introduction

Research on the use of polyethylene (PE) as a material for protective and anti-reflective (AR) coatings in photovoltaics (PV) has been conducted for a long time, but has intensified in recent years [1,2,3,4,5,6,7]. A simple method to fabricate moth-eye-like AR nanostructures in ion-beam etching onto polymer was presented in paper [8]. PE multilayer coatings can be used as an alternative to heavy glasses, as the light transmittance of these coatings was proven to be equal to that of high-quality solar glasses, amounting to 85–90%. Patterning and/or fabrication of small topological features in the coatings help to increase light trapping [9,10]. For example, an increase in energy efficiency of 1.34% for silicon modules with millimeter-sized cylindrical lenses obtained with a delicate embossing technique on an epoxy resin-based polymer (ERP) was as much as 10% for exposure with incidence angles ranging from 0 to 60° [11]. The authors of ref. [12] presented a hybrid concept where flat high-performance multi-junction solar cells made of III-V materials were mounted on the rear surface of concentrating PV solar modules. The results of external tests with two different designs of hybrid modules showed an absolute increase in the average daily efficiency ranging from 1.02% to 8.45%, depending on the weather conditions [12].

The authors of ref. [13] proposed a theoretical hybrid structure improving the efficiency of crystalline silicon solar cells: ordered nanoporous silicon (np-Si) with polythiophene (PT) filling inside the pores. This structure showed a significantly increased absorption coefficient as compared with that of ‘pure’ np-Si, proving np-Si/PT heterojunction to be a better light-absorbing material. The polymer that filled the pores produced a highly scattered valence band, which was the main route for hole transport. The Si/PT structure efficiently dissociated photo-induced electron-hole pairs and reduced the amount of material required for light absorption, leading to the fabrication of an inexpensive yet highly efficient solar cell with an efficiency of 30% for a material thickness of just 5 μm [13].

The authors of paper [14] managed to produce light-trapping structures on a silicon surface by various processing methods—laser, chemical and hybrid chemical/laser texturing. The modifications using various methods resulted in micro- and nanostructures that significantly reduced the reflectance of the silicon surface. Anti-reflective coatings for this type of cell could measurably enhance the light-trapping effect presented in [14]. The authors of refs. [15,16] confirmed that fillings or encapsulants may play a big role in PV cell efficiency, and that the physical properties of the materials used are very important. For example, the use of polyolefin elastomer (POE) or thermoplastic polyolefin (TPO) [15] instead of ethylene vinyl acetate (EVA) encapsulant in PV cells allowed for extending the spectral range to a 250–400 nm range (ultraviolet, UV range), which compensated for the low transmittance in the visible range. TPO showed a degree of crystallization three times higher than that with EVA, and with POE it was twice as high as that with EVA [15].

Increasing the efficiency of the cell, which is the subject of the research presented in this work, can be obtained by selecting an appropriate glass with high transmittance, close to 90%, and by the use of micro-concentrators (such as micrometer-sized lenses and/or sub-millimeter topological features [16]) with the functionality of an AR layer. In this paper, we present the results of the study of the effect of ‘micro-structuring’ of the PE coatings, achieved via formation of an array of micro air lenses with a diameter of 50 to 600 µm and micro-wrinkles enclosed between several layers of PE film, on the short-circuit current of silicon PV cells covered with glass.

## 2. Materials and Methods

The experiments were performed on the basis of Sec 4.4W 3bb crystalline silicon cell certified by the Fraunhofer Institute (ISE CalLab Cells, Fraunhofer, ISE, Freiburg Germany Multicrystalline silicon, Certificate Nr: 47058-PTB-12) with the following parameters: dimensions, 15.6 × 15.6 cm; short-circuit current *I*_sc_, 9.01903 A; maximum current *I*_max_, 8.46612 A; open circuit voltage *U*_oc_, 0.639647 V; maximum voltage *U*_max_, 0.5355 V; maximum power point MPP, 4.5336 W; fill factor FF, 78.5857%, efficiency ɳ, 18.6292%. The cell was covered with various commercially available (szklonawymiar.pl) glasses: greenhouse glass, float glass, optiwhite glass and acrylic glass (plexiglass). The transmittance measurements were performed using an AAA+ Quick Sun QS130CA solar radiation simulator (Endeas Oy, Espoo, Finland) [17] at Standard Test Conditions (STC: 1000 W/m^2^, 25 °C, AM1.5G). QS130CA includes a Toeliner TOE 8951/SN: 8009 current source and an Instek GDM-8246 current meter to determine *I*_sc_ short-circuit current, the most important solar cell parameter, with a precision better than 0.2%. 

An “open c-Si” module in the configuration being considered for study presented in this paper is shown in Figure 1. Such measurements were made for cells with covers directly laminated to the cell surface in the PV module manufacturing process. In our study, the optical properties of glass and glass–PE systems with a 30 mm air gap between the glass and the cell were determined. On the one hand, this introduced energy losses, which we had to neglect (and consider the effect of the gap as a systematic error), but on the other hand, this method allowed us to test the glass properties before the PV module was fully assembled, which significantly reduced the cost of testing. For that reason, UV lifetime tests were not conducted for this open panel configuration.

Two groups of glass types were tested, totaling 10 samples. These included: (i) 10 cm × 10 cm glass samples: 2 pieces of 2 mm thick float glass, 2 pieces of 4 mm thick float glass, 2 pieces of 4 mm thick greenhouse glass and 1 piece of optiwhite glass; and (ii), 16 cm × 16 cm glass samples: 1 piece of 1.5 mm thick acrylic glass (plexiglass), 1 piece of float glass and 1 piece of greenhouse glass, both 4 mm thick. As a result of determining the transmittance *T* of these samples on the basis of the current-voltage characteristics of the TSec4 cell covered with selected samples (Figure 2), three glass samples were selected with the highest transmittance. These were 2 mm thick float glass (*T*_Float_ = 89.6%), 4 mm thick optiwhite glass (*T*_Optiwhite_ = 90.0%,) and 1.5 mm thick acrylic glass (*T*_Plexi_ = 92.5%). The accuracy of determining the transmittance with this method was about 0.2%.

The photoconversion efficiency was determined on the basis of the standard method [18,19], which analyzes the change in *I*_sc_ of the covered cell with respect to *I*_sc_ of the ‘bare’ silicon cell.

The transmittance of the glasses in the wavelength range 175–3500 nm was double-checked using a CARY5000 spectrometer; these data are shown in Figure 3. At 555 nm wavelength, the obtained values were: *T*_Float_ = 91.07%, *T*_Optiwhite_ = 90.70% and *T*_Plexi_ = 92.16%. The accuracy of transmittance measurements was about 0.05%. The transmittances of the glasses determined with two methods appeared to be very similar.

Other glass samples, including float glass with a thickness of 4 mm and greenhouse glass, showed photoconversion efficiency below 85% and were excluded from further consideration.

## 3. Results

Fifty-seven (57) types of coatings were made with two, four and six layers of PE films with thicknesses of 9, 12 and 23 µm on 10 × 10 cm glass samples (float and optiwhite glass) and 16 cm × 16 cm acrylic glass samples. The density of the PE films was 0.918–0.920 g/cm^3^. Each PE film covered just 41% of the crystalline silicon cell, so the rest of the cell was exposed and the short-circuit current component from the exposed part of the cell was subtracted from the value of the total *I*_sc_ registered with a QS130CA simulator. The films were commercially available (dobrafolia.pl, bifol.pl, etc.) PE films, and they were applied by hand, ensuring that there were no traces of material tension in the form of excessively large wrinkles, which was checked with an optical microscope. On the other hand, smaller-sized wrinkles help to enclose air bubbles between the glass and films, and these bubbles served as micro air lenses. 

Visualization of the structure of microbubbles and micro-wrinkles encapsulated between the acrylic glass and two layers of 12 um thick film for two randomly selected areas of 2.5 mm × 2.5 mm was performed. Imaging and sizing, presented in Figure 4, were performed using a 3D laser scanning microscopy OLS5100. Three types of typical wrinkle widths were observed: D1, with a ‘width’ (diameter) less than 50 µm; D2, with a diameter of 50–600 µm; and D3, with a diameter of 600–1200 µm. Additionally, two types of wrinkle length were observed: L1, with a length of several millimeters, and L2, with a length of several centimeters. Each system was prepared anew three or four times until the width (‘diameter’) of the wrinkles did not exceed 600 µm and the surface density of the air bubbles and wrinkles did not exceed, 15%. Wrinkle density was calculated for 5–7 averaged randomly selected fields sized 2.5 mm × 2.5 mm.

Based on the current-voltage characteristics shown in Figure 5 for acrylic glass, the photoconversion efficiency of the cells covered with the studied ‘glass-PE films’ structures were determined, and the data obtained are summarized in Table 1.

In this Table, column 1 lists the object, for which the photoconversion efficiency was determined; *d* is the thickness of the glass, and the size of the glass sample is also given. PE films were applied to the glass on both sides. Column 2 shows the value of *I*_sc_ measured experimentally; the values obtained were reduced to the area of the samples, so the comparison of the results obtained on different samples was correct. The systems with the simplest coatings, 2 × 9 µm thick films, were chosen as a ‘reference coating’ (RC), and the value of the current gain was calculated for the remaining films against the *I*_sc_ of these coatings. The final column in Table 1 shows the *I*_sc_ gain; the accuracy of determination of the gain was 0.02%. For the samples where no gain was achieved, the cells in this column were left empty.

On the basis of the analysis of the data in Table 1, it can be observed that the most substantial *I*_sc_ gain was achieved for 1.5 mm thick acrylic glass; in relation to RC, it was:with two layers of 12 µm thick PE films—4.05%;with two layers of 23 µm thick PE films—1.8%;with four layers of 12 µm thick films—1.34%.

The 2 mm thick float glass with two 12 µm thick PE films showed a current gain of only 0.06%, and the optiwhite glass showed a gain of 0.09% for the glass with two 12 µm thick films.

Thus, the system with acrylic glass and two layers of PE film with a thickness of 12 µm each appeared to be the most effective coating for the PV cells.

For the studied system, which was placed directly on the silicon cell, the total light reflection averaged over three different systems with two layers of 12 µm thick PE films was measured; the results are shown in Figure 6. The light reflection was also recorded for the silicon cell and the silicon cell covered with acrylic glass. The same Tsec4 crystalline silicon cell was used here to determine the I–V characteristics, shown in Figure 5. The thickness of this cell is 200 µm, and its surface was coated with an anti-reflecting acrylic silicon nitride in the fabrication process; three silver busbars with a width of 1.5 mm were placed on it, and the distance between them was 52 mm. The commercial acrylic glass material does not have anti-reflective properties, while the investigated system, PE film–glass–PE film together with air bubbles and wrinkles enclosed between the PE films and the glass, showed anti-reflective coating properties, which were manifested by a 0.5–4.6% total reflection decrease in the ranges 275–375 nm and 600–1200 nm compared with the reflectance spectrum of silicon cell under plexi (shown in Figure 6).

## 4. Discussion

During the application and slight stretching of the film (performed just for good adherence to the smooth, non-structured surface of the glass), air bubbles with diameters 50–600 µm were formed, as discussed above. While the nominal thickness of a 1.5 mm thick glass with two layers of 12 µm thick PE films should equal 1.524 mm, the thickness of the PE layer was actually increased by the height of the wrinkles and air bubbles to values ranging from 1.550 mm to 1.850 mm, as was determined using a digital optical microscope with a magnification of ×50 with the accuracy of the determination of the thickness Δ*d* = 0.002 mm. The resulting structure with the wrinkles and the bubbles resembled a well-known ‘grooves’-like glass texture, which is used for enhancing the PV modules’ performance. A good example of this texture was presented by the authors of ref. [20]. The similarity between this texture and the structure formed in our samples is especially close in the case of cross-overlapping of successively applied PE films.

The ‘grooves’-like texture is most effective for the angles of incidence of solar radiation in the range of 55–80°; at these angles, it showed an open current gain of as much as 60%, while for normal incidence, only 1.4% of the gain was obtained for the same structure [20] (see Figure 7, dark bars).

To compare the results obtained in this work with those obtained for the ‘grooves’-like texture, simulations for the most effective system (acrylic glass with two layers of 12 µm thick PE film) were performed. In the simulation, the nature of *I*_sc_ gain amplification seems to be similar to that presented in ref. [20]. The dependence of the gain on the incidence angle was described by a polynomial of the fifth order:*y* = 4 × 10^−7^*x*^5^− 7 × 10^−5^*x*^4^ + 4 × 10^−3^*x*^3^ − 0.097*x*^2^+ 0.777 *x* + 1.099,(1)
where *x* is the angle, *y* is the *I*_sc_ gain and the initial condition for the normal angle of incidence is that the *I*_sc_ gain is equal to 1.4% (indicated by black bars in Figure 7).

In modeling this process for *I*_sc_ gain (0°) = 4.05% presented here (white bars in Figure 7), a nearly 50% reduction in final *I*_sc_ gain as a function of incident angle was accounted. These losses were primarily due to the scattering of solar radiation on film wrinkles and air lenses with sizes smaller than 25% of their maximum size. Comparing the properties of two coats shown in Figure 7, we find that despite a significant theoretically assumed reduction in *I*_sc_ gain for the system of acrylic glass and two layers of 12 µm thick PE films, in reality, its capabilities appear to be at least 30% greater than for ‘grooves’-like textured glass.

The results obtained in the present research are indicative of the fact that a high value of the current gain for acrylic glass coated with PE films can be achieved. Acrylic glass, or plexiglass, belongs to the same group of materials as PE films; they have exactly the same heat transfer coefficient. In addition, acrylic glass had the highest transmittance of all glasses studied. Therefore, the best results obtained for this type of glass can be explained by the fact that when applying PE films to acrylic glass, we dealt with the same family/group of polymeric materials. For the best configuration found in this work, a gain of ~4% was achieved, and for incidence angles in the range of 70–80° it should certainly be much greater than 4%, possibly even more than the 60% reported by authors of ref. [20] for ‘grooves’-textured glass.

**Figure 7 materials-16-04004-f007:**
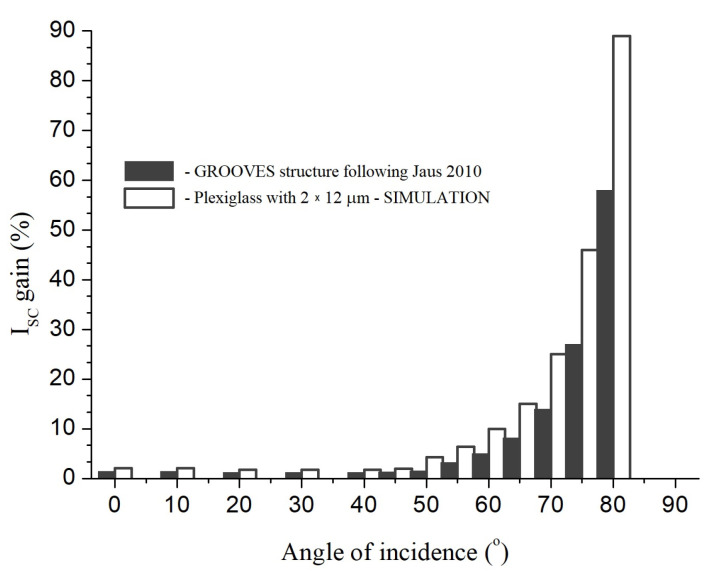
Relative *I*_SC_ gain as a function of angle of incidence for ‘grooves’-like texture presented in ref. [20] (dark bars) and the results of simulation for acrylic glass with two 12 µm thick PE films (white bars).

The results are consistent with reports of using a 30 µm thick PE layer to improve lifetime, but with a significant loss of the energy efficiency of silicon crystalline cells, over which a single PE layer was placed, and the next encapsulated EVA [21]. The method of improving the efficiency of silicon cells described in ref. [13] could also be applied to PE films, and the use of a different encapsulant should extend the spectral range of solar radiation converted into electricity in the modules modified by the addition of PE films.

In regard to future research directions, we may also suggest that further efficiency enhancement may be achieved by using specialized glasses, such as a low-iron AR-nanostructured glass manufactured by Effect Glass [22]. Improvement of the efficiency of PV elements covered with this glass in the spectral range of 700–800 nm at the incidence angles 70–80° was reported to be of the order of at least 3% [22]. In our own research [19], an *I*_sc_ gain of 2% was obtained for a textured glass with an AR layer. In the experiments discussed in ref. [19], a sample of unstructured glass measuring 10 cm × 10 cm and with thickness 3.2 mm showed a transmittance (measured with the use of QS130CA with the same TSec4 reference cell as the one used in current work) equal to 82.26%. In this work, the best coating, consisting of acrylic glass with 2 × 12 µm thick PE films, showed a transmittance of 86.38%. Therefore, using the specialized AR glass should clearly enhance the short-circuit current even more. Optimizing the ‘glass-PE’ configuration and fabricating a mechanically robust composite glass–polymer system should allow for further improving the photoconversion capabilities of silicon PV modules.

## 5. Conclusions

In this paper, the effect of various PE coatings on the short-circuit current of silicon PV cells covered with different types of glasses was studied. The best current enhancement of 4.05% was achieved for the coating consisting of a 1.5 mm thick acrylic glass with 2 × 12 µm thick PE films. 

It is believed that this effect is due to the formation of an array of micro air lenses with an average diameter of 50 to 600 µm and micro-wrinkles enclosed between the layers of the film and at the film/glass interface. 

A similar result was achieved in paper [23], where efficient light trapping in vertically aligned n-type and p-type 2D random (interconnected pores) array of Si-NWs (NanoWireS) was reported, and antireflection properties of these synthesized Si-NWs showed maximum light trapping because of the high density of NWs. The power cell efficiency of Si-NWs exhibited about 4%, but technological requirements and research costs were many times higher than those presented in this paper.

The study of light trapping in polyethylene protective coating with anti-reflective properties for silicon photovoltaic cells has been explored to add new dimension to the field of photovoltaic application. 

## Figures and Tables

**Figure 1 materials-16-04004-f001:**
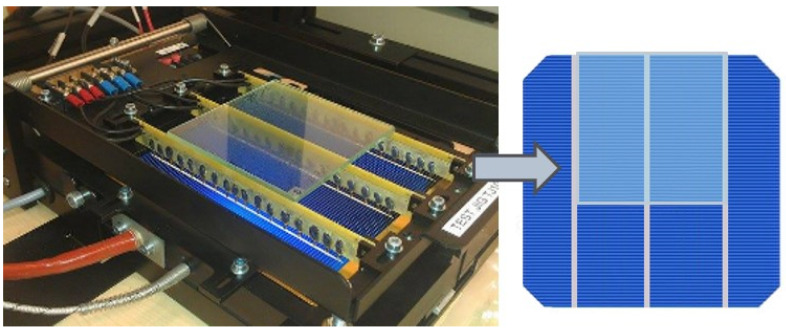
An “open c-Si” module in the configuration being considered for study presented in this paper. Picture shows part of QS130CA equipment.

**Figure 2 materials-16-04004-f002:**
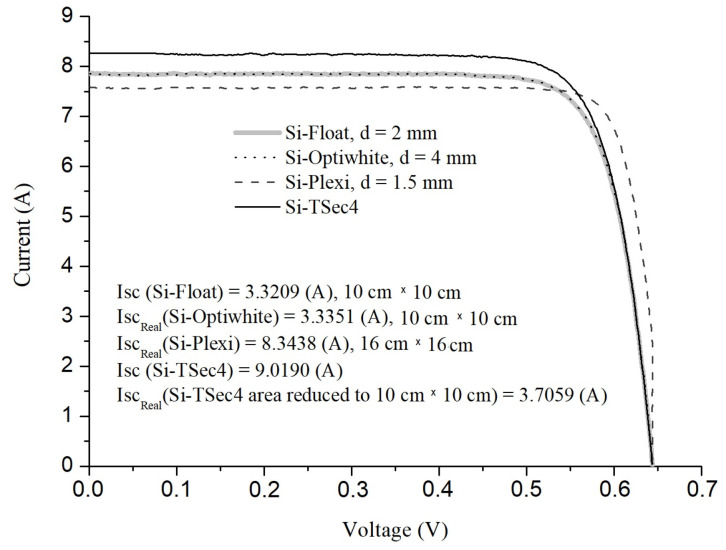
Current-voltage characteristics of PV cells covered with glasses used for the determination of the transmittance of the glasses with the use of a QS130CA solar radiation simulator.

**Figure 3 materials-16-04004-f003:**
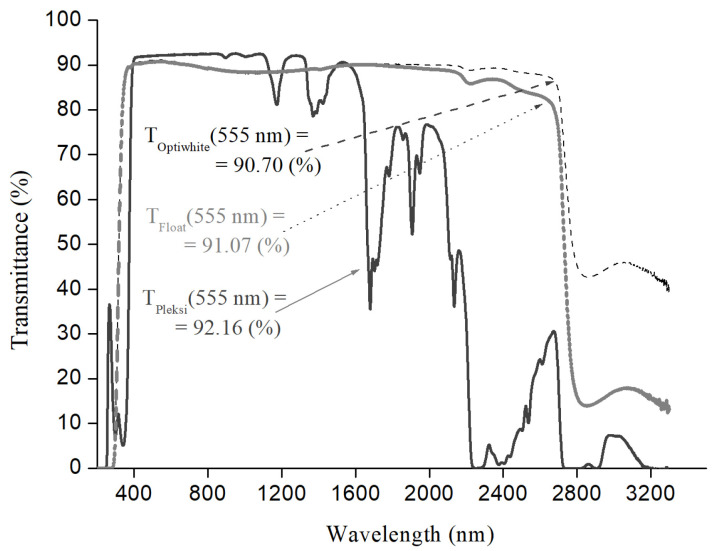
Transmittance of selected glasses determined with the use of a CARY5000 integrating sphere spectrophotometer.

**Figure 4 materials-16-04004-f004:**
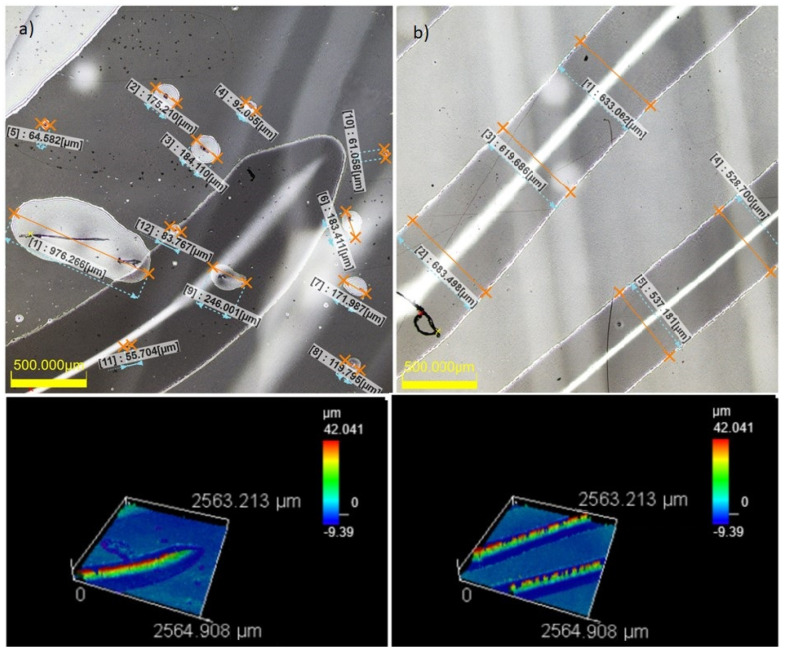
Picture of air bubbles and wrinkles for two randomly selected two different areas (**a**) and (**b**) of acrylic glass with 12 µm width polyethylene film on top and bottom sides of sample. Bottom pictures showthe height of wrinkles, upper part of picture-diameters of air bubbles and wrinkles.

**Figure 5 materials-16-04004-f005:**
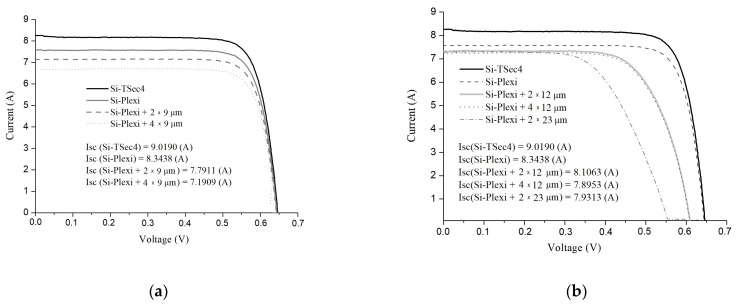
Current-voltage characteristics of PV cells covered with ‘acrylic glass-PE film’ structures with 9 µm thick PE films (**a**) and with 12 µm and 23 µm thick films (**b**), used for the determination of photoconversion efficiency of the cells.

**Figure 6 materials-16-04004-f006:**
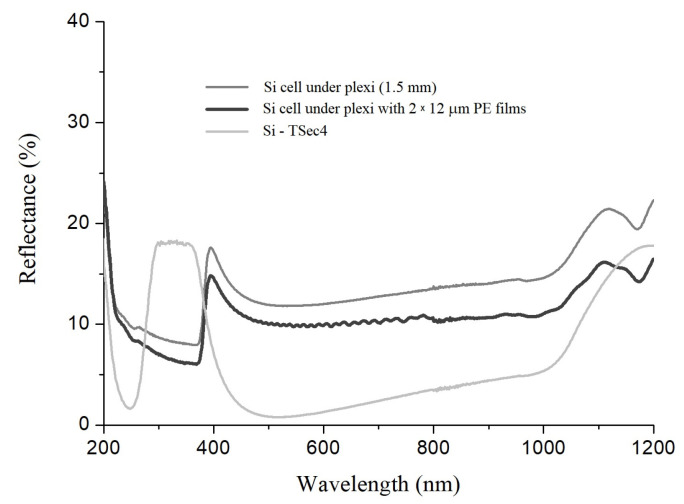
Total light reflection for silicon cell, silicon covered by plexiglass and silicon covered by investigated optimal system. Spectra registered on Cary5000.

**Table 1 materials-16-04004-t001:** Photoconversion efficiency of the studied systems, determined with the Solar Radiation Simulator QS130CA.

Studied Object	PE Film Thickness, µm	*I*_sc_, A	Efficiency of Photoconversion, %	*I*_sc_ Gain, %
Float glass, *d* = 2 mm, 10 × 10 cm	Pure glass	8.6340	89.61	
	2 × 9	8.4917	85.77	RC
	4 × 9	8.3656	82.37	
	6 × 9	8.1033	75.29	
	2 × 12	8.4937	85.82	0.06
	4 × 12	8.2426	79.05	
Optiwhite glass, *d* = 4 mm, 10 × 10 cm	Pure glass	8.6483	90.00	
	2 × 9	8.4909	85.75	RC
	4 × 9	8.2787	80.02	
	6 × 9	7.8965	69.71	
	2 × 12	8.4664	85.82	0.09
	4 × 12	8.2679	79.05	
	2 × 23	8.2352	78.85	
	4 × 23	7.9551	71.29	
Acrylic glass, *d* = 1.5 mm, 16 × 16 cm	Pure glass	8.3438	92.51	
	2 × 9	7.7911	86.38	
	4 × 9	7.1909	79.73	
	6 × 9	6.7526	74.87	
	2 × 12	8.1063	89.88	4.05
	4 × 12	7.8953	87.54	1.34
	2 × 23	7.9313	87.94	1.80
	4 × 23	7.0493	78.16	

## Data Availability

The data presented in this study are available on request from the author.

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
