# Peer review of "Polyethylene Protective Coating with Anti-Reflective Properties for Silicon Photovoltaic Cells"

_materials, 2023, doi:10.3390/ma16114004_

Round 1
Reviewer 1 Report
The article "Polyethylene Protective Coating with Anti-Reflective Properties for Silicon Photovoltaic Cells" by Malgorzata Pociask-Bialy on the influence of polyethylene coatings on silicon photovoltaic cells is highly relevant and new. In light of the growing demand for sustainable energy and the need to improve the efficiency of solar panels, the search for new ways to improve their performance is of utmost importance. The study is focused on finding different combinations of coatings and glasses, and identifying a specific coating formula that provides the best current increase. The assumption of the formation of micro-wrinkles and air bubbles, enhancing light capture, is a bright result that can be useful for future research and development in the field of solar energy.
Before recommending the article for publication, it would be desirable for the author to comment on the following:
1. It would be extremely helpful to provide the diffuse reflection coefficient over a wide range of wavelengths (as shown, for example, in https://www.mdpi.com/1996-1944/16/6/2350/htm).
2. The assumption about "micro-wrinkles" would be confirmed if the structure were demonstrated using SEM.
3. How long-lasting is such coating? It is desirable for the solar panel to last for 7 years or more.
4. It would also be useful to schematically represent the solar panel in the configuration being considered.
Reviewer 2 Report
The manuscript aims to investigate the impact of polyethylene (PE) coatings on the short-circuit current of glass-covered silicon photovoltaic cells in order to enhance the short-circuit current. The authors demonstrate that the best current gain of 4.05% is attained for the coating that combines acrylic glass with PE films. This result is attributed to the development of an array of micro-wrinkles and micrometer-sized air bubbles in the films that act as micro-lenses and enhance light trapping.
Between lines 97 and 113, modify the area notation to "10 × 10 cm²" instead of "10 × 10 cm."
It is crucial to provide microscope images and describe how the wrinkles' diameters and lengths were measured.
It is unclear whether the solar simulator includes a source meter for determining the short-circuit current. It would be beneficial to provide more information on how the measurement was taken.
The author refers to an unpublished article. It is recommended that the cited articles have gone through peer review before being included in the journal Materials.
After implementing these revisions, the manuscript should be suitable for publication in the journal Materials. The preprint of the submitted article is available for download at the following link: https://www.preprints.org/manuscript/202304.1010/v1/download.
Moderate editing of English language
Reviewer 3 Report
The manuscript by Malgorzata Pociask-Bialy deals with the investigation of different commercial polymeric films to be used as Anti Reflective Coatings in conventional photovoltaic application (i.e. as coatings for Si-based devices).
The organization of the manuscript is good, albeit some more recent references could be added, especially the ones related to the importance of improve the light utilization efficiency of silicon panels.
Results are clearly presented and properly discussed.
It would be interesting to add a section to monitor the behaviour of the different coatings under accelerated degradation tests.
Overall, the manuscript could be accepted after minor revisions.
Round 2
Reviewer 1 Report
The authors answered all questions. The article may be published.
Reviewer 2 Report
After revisions have been made, I recommend publishing the manuscript in the journal Materials.
Minor editing of English language required